# Accuracy of Dental Implant Placement with Dynamic Navigation—Investigation of the Influence of Two Different Optical Reference Systems: A Randomized Clinical Trial

**DOI:** 10.3390/bioengineering11020155

**Published:** 2024-02-04

**Authors:** Anne Knipper, Katharina Kuhn, Ralph G. Luthardt, Sigmar Schnutenhaus

**Affiliations:** 1Center for Dentistry, Dr. Schnutenhaus Community Health Center (CHC) GmbH, Breiter Wasmen 10, 78247 Hilzingen, Germany; knipper@schnutenhaus.de; 2Department for Dentistry, Clinic for Prosthodontics, Ulm University, Albert-Einstein-Allee 11, 89081 Ulm, Germany; katharina.kuhn@uniklinik-ulm.de (K.K.); ralph.luthardt@uniklinik-ulm.de (R.G.L.)

**Keywords:** dental implants, computer-aided design, computer-aided manufacturing

## Abstract

This randomized prospective clinical study aims to analyze the differences between the computer-assisted planned implant position and the clinically realized implant position using dynamic navigation. In the randomized prospective clinical study, 30 patients were recruited, of whom 27 could receive an implant (BLT, Straumann Institut AG, Basel, Switzerland) using a dynamic computer-assisted approach. Patients with at least six teeth in their jaws to be implanted were included in the study. Digital planning was performed using cone beam tomography imaging, and the visualization of the actual situation was carried out using an intraoral scan. Two different workflows with differently prepared reference markers were performed with 15 patients per group. The actual clinically achieved implant position was recorded with scan bodies fixed to the implants and an intraoral scan. The deviations between the planned and realized implant positions were recorded using evaluation software. The clinical examinations revealed no significant differences between procedures A and B in the mesiodistal, buccolingual and apicocoronal directions. For the mean angular deviation, group B showed a significantly more accurate value of 2.7° (95% CI 1.6–3.9°) than group A, with a value of 6.3° (95% CI 4.0–8.7°). The mean 3D deviation at the implant shoulder was 2.35 mm for workflow A (95% CI 1.92–2.78 mm) and 1.62 mm for workflow B (95% CI 1.2–2.05 mm). Workflow B also showed significantly higher accuracy in this respect. Similar values were determined at the implant apex. The clinical examination shows that sufficiently accurate implant placement is possible with the dynamic navigation system used here. The use of different workflows sometimes resulted in significantly different accuracy results. The data of the present study are comparable with the published findings of other static and dynamic navigation procedures.

## 1. Introduction

In the course of implant planning, in addition to the obligatory protection of neighboring structures, such as adjacent teeth, nerves or the maxillary sinuses, aspects of subsequent prosthetic restoration should also be taken into account. This means that the position and height of the implant should be set in such a way that prosthetic restoration is possible without any problems and provides a functionally and esthetically perfect result. The prosthetically correct implant position also has an influence on the long-term success of the implant prosthetic restoration, for example, through the hygiene capability of the prosthesis [1]. The optimal use of the available bone is essential for long-term success [2]. This forward-thinking planning only fulfills its purpose if it can be implemented accordingly in the patient’s mouth.

In order to be able to implement this predictably, a digital approach is increasingly being used in addition to the conventional analog procedure. Three-dimensional imaging, an intraoral digital surface data set of the clinical situation and prosthetic planning serve as the basis for optimal planning. In this way, implant planning can take place based on the premise of the subsequent prosthetic restoration [3].

Implants can be inserted in different ways. A distinction can be made between freehand implant placement, implant placement using insertion guides, i.e., statically navigated implant placement, and dynamically navigated implant placement. Many studies show higher accuracy for navigated implant placement compared to the conventional freehand method. The computer-assisted approach is considered safer and faster and leads to more predictable results [4]. Statically navigated implant placement is divided into fully guided implant placement, half-guided implant placement, which includes drill guides and pilot drill guides, and non-computerized approaches. The highest accuracy is achieved with fully guided implant placement in conjunction with a flapless approach and a tooth-supported insertion template [5].

With statically navigated implant placement, the desired implant position is determined in advance using virtual three-dimensional planning software. The functional and esthetic requirements for subsequent prosthetic restoration are taken into account. The virtual planning is transferred to the surgical site using computer-aided design/computer-aided manufacturing (CAD/CAM) of surgical guides. The virtual procedure, also known as backward planning, ensures the ideal positioning of the implant and leads to a more controlled and safer implantation as well as a predictable prosthetic result. 

The use of drilling templates to transfer the planning has already been studied in detail and leads to predictable results [5,6]. There are many studies that have investigated the accuracy of static navigation and the possible factors influencing it. The intraoral positioning of implant drill guides and their fixation have a decisive influence on the accuracy [7]. Tooth-supported drill guides lead to a more precise result than mucosa- or bone-supported drill guides [8,9].

In addition to the different materials used for surgical guides, the manufacturing process can also influence the accuracy [10,11,12]. Drill guides can be laboratory-made, milled or printed [13]. Several studies have shown that the use of different drill sleeves has an effect on the accuracy of an implantation. Different drill sleeve heights lead to possible deviations in accuracy [14]. It is also possible to use drill guides with or without metal sleeves. In the variant without metal drill sleeves, the template can be open or closed. All the variants mentioned have an influence on the accuracy [15]. Application errors or the process itself often lead to greater inaccuracies [16]. The jaw to be implanted and the implant position itself have a decisive influence on accuracy [9].

When insertion guides are used, significantly more precise results are achieved in terms of transfer accuracy and achieving the desired implant position compared to freehand implant placement [8,9,10,11].

A disadvantage of fully guided static navigation is the fact that no condition-related changes can be made intraoperatively, i.e., the original planning cannot be modified intraoperatively [5].

As an alternative to static computer-assisted implant placement, there are dynamic procedures that also enable navigated implant placement [17]. During the preparation of the implant site and implant placement, the surgeon navigates in real time using a three-dimensional representation of the actual situation and the planning on a screen. With the help of optical tracking systems and reference markers, the positions of the instruments are recognized and displayed on the screen [18].

The accuracy of various dynamic implantation procedures has been investigated in mainly preclinical studies. These results need to be further confirmed in clinical studies [19,20,21]. Due to high costs and a complicated clinical setup, they have not yet been widely used clinically. In principle, dynamically navigated implantation can achieve similarly accurate results as statically navigated implantation. However, these two procedures show significantly more accurate results than freehand implantation [18].

Dynamic computer-assisted implantation procedures have many advantages, such as (for example) the open-sourced system, which means that any implantation system can be used. In addition, the disadvantages of other implantation procedures can be eliminated. The storage of static surgical guides and their susceptibility to inaccuracies are eliminated, especially with a flapless approach [17]. Furthermore, the surgical field can be viewed at all times as the surgical guides are no longer required. This means that the implantation remains controllable, i.e., the original planning can be modified intraoperatively if necessary [5]. Another advantage is that, in contrast to surgical guides, the procedure can also be used when vertical space is limited [22]. The learning curve of the procedure must be taken into account, as the entire team, especially the surgeon, needs to be trained due to the high complexity of the procedure [23]. However, it has been found that there are no significant differences in accuracy when the implantation is performed by an experienced surgeon or a novice surgeon after a training period has taken place [24,25,26].

The mouth opening, possible movements of the patient and the possibly restricted view of the surgical field in the posterior region can have an influence on the accuracy [27]. The material of the reference markers used can lead to artifacts and thus have an influence on the accuracy [28] as well as the positioning of the fiducial markers themselves [29]. Another possible source of inaccuracies are the individual steps of the digital workflow, which can add up [11]. The quality of patient care should be improved with the further development of implantation procedures based on virtual reality and augmented reality [30].

A comparison of various studies on dynamically navigated implant placement shows that the coronal and apical deviations, as well as the angular deviations, are highly dependent on the system used. The studies generally show that clinically sufficient accuracy can be achieved [31]. A comparison of studies on static navigation shows it has similar accuracy to dynamically navigated implantation [26,32,33]. Marques-Guasch et al. conclude in their meta-analysis that dynamic navigation is a valid alternative treatment that achieves similar accuracy to static navigation [34]. The accuracy of freehand implantation is clearly inferior to that of the other two methods [35,36]. Overall, only a few clinical studies on the accuracy of dynamically navigated implantation are available to date. The studies differ in the use of the respective navigation system, the optical tracking system and the position of the reference markers. In addition, different implant planning systems and implants were used. This heterogeneity does not lead to a well-founded assessment of clinical applicability [32].

The aim of the present randomized prospective clinical studies was to evaluate the transfer accuracy of fully navigated implantations using a dynamically navigated implantation system, Mininavident (Mininavident, Liestal, Switzerland). In the system examined here, an intraorally inserted marker is used. The marker can be integrated and designed in the planning software either fully digitally or with prefabricated auxiliary parts. The evaluation took into account the different procedures used to collect the planning data sets.

It was hypothesized that the use of different workflows has no influence on the accuracy.

## 2. Material and Methods

### 2.1. Study Structure

This randomized, controlled clinical trial (RCT) was approved by the Ethics Committee of the Baden-Württemberg Medical Association under the number F-2020-113-z on 20 November 2020. It has been registered in the German Clinical Trials Register and the International Clinical Trials Registry Platform by the WHO under DRKS00023691.

The inclusion criteria for this study were as follows:Interocclusal gaps in the maxillary and/or mandibular posterior regions.Free-end situations in the maxillary and/or mandibular posterior regions.The extraction must have taken place at least 3 months ago. Only early implantations (3–6 months post-extraction) and late implantations (>6 months post-extraction) are performed.The marker tray used as a reference point must be tooth-supported. Therefore, there must be at least 6 remaining teeth in the jaw to be implanted.The patient must demonstrate good oral hygiene and compliance.A pre-implant hygiene phase and pre-treatments must be completed.The patient’s declaration of consent must be available.

The exclusion criteria were as follows:Persons under 18 or persons without legal capacity.Untreated acute periodontitis with pocket depths > 4 mm.Heavy smokers (more than 10 cigarettes/day).Taking bisphosphonates.Pregnant women.Alcohol and drug addicts.Patients with an infectious disease such as hepatitis or HIV/AIDS.Patients with severe diabetes mellitus.Immediate implantations.Width of the gap in the mesiodistal direction for single-tooth gaps < 7.1 mm.

A total of 30 patients were included in the study, who were divided into 2 groups. The groups were to represent 2 different workflows (Figure 1), A and B, each with 15 patients. 

### 2.2. Workflow A

In workflow A, a CBCT (PaX-i3D Greenxt, orangedental GmbH, Biberach, Germany) of the patient was taken first. All CBCTs were performed with a resolution of 0.2 voxels and an acquisition time of 9 s. The marker of the DENACam system (Mininavident, Liestal, Switzerland) (Figure 2) required for the subsequent implantation was then placed on the contralateral side of the patient’s mouth. The marker was placed using a DENATray (Mininavident, Liestal, Switzerland) and DENABeads (Mininavident, Liestal, Switzerland), a thermal impression material. The impression material was placed in a container of boiling water, and after about 20 s, the resin became plastic and transparent. The impression material could then be removed and pressed into the tray. The tray was fixed on the intended teeth and not moved until the plastic had hardened. The tray was then removed and repositioned to check that it could not be moved. The marker could now be attached to the tray. An intraoral scan (Trios 3 Scanner, 3Shape A/S, Copenhagen, Denmark) was taken of the jaw to be treated, both with and without the tray in place. The data sets of the CBCTs and the scans were imported into the planning software coDiagnostix (Version 9.11, Dental Wings GmbH, Chemnitz, Germany) and superimposed. The implant was planned taking into account the anatomical structures, the neighboring teeth and subsequent prosthetic restoration. The planning data set was exported to a USB stick and transferred to the DENACam system (Mininavident, Liestal, Switzerland). For implant placement, the marker tray was again placed in the patient’s mouth, and the implant was inserted according to a prescribed drilling protocol using an implant motor (iChiropro, Bien Air Dental, Biel, Switzerland) (Figure 3, Figure 4 and Figure 5). All implantations were performed by an experienced dentist.

### 2.3. Workflow B

For workflow B, a CBCT and an intraoral scan of the patient were taken. The data sets were superimposed in the planning software, and a marker tray was designed on the contralateral side using the software, which was then printed using a 3D printer (Formlabs Version 3, Formlabs Inc., Somerville, MA, USA). The implant was planned as described. For implant placement, the printed tray with a marker was placed in the patient’s mouth, and the procedure was continued as described in workflow A. In this workflow, all implantations were also performed by an experienced dentist.

### 2.4. Randomization

Randomization was carried out using 30 envelopes, each containing a sheet of paper. Each envelope contained a group, A or B, and a number from 1 to 15. Each combination occurred only once. The envelopes were drawn blind and handed to the operator.

### 2.5. Registration of the Implant Position

After the insertion of the implants, an intraoral scan was made before subsequent closure of the implants using cover screws or gingiva formers and wound care. For this purpose, the corresponding scanbodies (NC or RC monoscan bodies, Straumann Institut AG, Basel, Switzerland) were screwed onto the implants. This surface data set in STL format was overlaid with the planning data set for evaluation. Superimposition and evaluation were performed by a dentist who was neither involved in the planning nor in the implantation. The automated surface that best fit the iterative closest point algorithm in the treatment evaluation mode of the coDiagnostiX V 9.11 (Dental Wings GmbH, Chemnitz, Germany) software was used to overlay the preoperative planning data set with the postoperative digital data set. Several matching points of the respective data sets were selected and then superimposed by the software. The overlay was checked for accuracy and corrected manually if necessary. In the coDiagnostiX software, the respective implant could then be optically assigned via a defined point on the scan body. For this purpose, the corresponding implant was selected in advance from the implant database. The planned implant was displayed with the actually inserted implant, and evaluation parameters were created by the software.

The following measurements were taken for the metric analysis of the implant positions (Figure 6):Three-dimensional deviation: the three-dimensional deviation of the center points between the planning and the clinically achieved positions of the implants measured at the implant shoulder and the implant apex (corresponds to the Euclidean distance);Apicocoronal deviation: includes the difference in height, i.e., the spatial offset in the vertical direction, measured at the center of the implant shoulder;Axial deviation: angular deviation of the implant axes from the implant planning and clinically achieved implant positions;Two-dimensional deviation: the two-dimensional deviation in the mesiodistal and buccolingual directions measured at the implant shoulder and the implant axis.

### 2.6. Sample Size

As no clinical results of a similar study were available, no biometric estimation of the sample size could be performed. Our own preclinical studies on the procedure showed that significant differences were detected with the selected sample size. The present study should therefore be classified as a pilot study with 15 patients in each test group.

### 2.7. Statistical Methods

The mean value, the standard deviation, the 95% confidence interval and the minimum and maximum values were calculated for all the variables determined. After testing for a normal distribution using the Shapiro–Wilk test, a statistical test for group differences was carried out. A global test of the groups was carried out depending on the determination of a normal distribution (analysis of variance (ANOVA) or Kruskal–Wallis). If there were significant differences between the groups (*p* < 0.05), pairwise comparisons were applied using post hoc tests.

## 3. Result

A total of 30 patients were recruited for the clinical part. In the end, three cases could not be included in the evaluation. In three cases of group A, the reference marker could not be recognized during the operation, and the implantation could not be dynamically navigated. 

Overall, 27 implantations in 27 different patients could therefore be included in the analysis (Table 1 and Table 2).

The mean angular deviation for group A was 6.3° (95% CI 4.0–8.7°), the minimum was 0.4° and the maximum was 13.9°. Group B showed a mean value of 2.7° (95% CI 1.6–3.9°) for the angular deviation, with a minimum of 0.7° and a maximum of 8.3°. In comparison, group B showed significantly more accurate values than group A (Figure 7).

The mean 3D deviation at the implant shoulder for workflow A is 2.35 mm (95% CI 1.92–2.78 mm). A minimum value of 1.31 mm and a maximum value of 3.50 mm were determined. Workflow B showed an average 3D deviation of 1.62 mm (95% CI 1.2–2.05 mm), with a minimum of 0.22 mm and a maximum of 2.99 mm. This showed significantly higher accuracy for procedure B (Figure 8).

Similar values could be determined for the 3D deviation at the implant apex. Procedure A showed a mean value of 3.0 mm (95% CI 2.3–3.7 mm), a minimum of 1.7 mm and a maximum of 4.7 mm, which is significantly less accurate than the mean value of procedure B with a value of 1.8 mm (95% CI 1.3–2.3 mm), a minimum of 0.6 mm and a maximum of 4.0 mm (Figure 9).

The 2D deviation at the implant apex and shoulder was also significantly more accurate in group B than in group A. 

In the mesiodistal direction, procedure A showed a mean deviation of 0.17 mm (95% CI −0.9–1.2 mm) at the implant shoulder, with a minimum value of −2.8 mm and a maximum value of 2.7 mm. For procedure B, a mean deviation of 0.6 mm (95% CI 0.2–1.04 mm) was determined in comparison, with a minimum value of −0.6 mm and a maximum value of 2.4 mm. At the implant apex, the mean deviation in the mesiodistal direction in group A was 0.1 mm (95% CI −1.6–1.7 mm), with a minimum of −3.4 mm and a maximum of 4.3 mm. Group B showed a mean deviation of 0.7 mm (95% CI 0.1–1.24 mm). The minimum was −0.7 mm, and the maximum was 3.3 mm.

Workflow A showed a mean deviation of −0.1 mm at the implant shoulder in the buccolingual direction (95% CI −0.8–0.5 mm), with a minimum value of −1.3 mm and a maximum of 1.8 mm. At the apex, the mean deviation was −0.3 mm (95% CI −1.2–0.7 mm). The minimum was −2.9 mm, and the maximum was 2.2 mm. In comparison, workflow B showed a mean deviation at the implant shoulder in the buccolingual direction of −0.5 mm (95% CI −0.93–−0.1 mm), with a minimum value of −2.0 mm and a maximum value of 0.9 mm. At the implant apex, a mean deviation of −0.6 mm (95% CI −1.1–−0.14 mm) was found, with a minimum value of −2.1 mm and a maximum value of 0.7 mm.

The mean deviation in the apicocoronal direction for procedure A was −1.34 mm (95% CI −1.9–−0.8 mm) at the implant shoulder and −1.3 mm (95% CI −1.8–−0.7 mm) at the implant apex. Minimum and maximum values of −2.3 mm and 1.0 mm at the implant shoulder and −2.3 mm and 1.0 mm at the implant apex were measured. Procedure B showed an average deviation of −0.74 mm (95% CI −1.3–−0.2 mm) in the apicocoronal direction at the implant shoulder, as well as a minimum of −2.1 mm and a maximum of 0.8 mm. At the implant apex, the mean deviation was −0.7 mm (95% CI −1.3–−0.2 mm), with a minimum of −2.1 mm and a maximum of 0.8 mm.

No significant differences were found between procedures A and B in the mesiodistal, buccolingual and apicocoronal directions (Table 3).

## 4. Discussion

The aim of implant prosthetic restoration is to replace missing teeth and restore the masticatory system while taking functional and esthetic aspects into account. The basis for a perfect prosthetic restoration is optimal implant planning from a prosthetic point of view with ideal positioning of the implants. Various computer-aided procedures have been established to transfer this goal optimally to the patient’s mouth. The accuracy of the implementation in all guided implantation procedures is influenced by numerous factors. Contrary to the hypothesis put forward in this study, it has been shown that even the manufacturing process of the marker holder required in this system has a significant influence on the accuracy.

Compared to static navigation, relatively few clinical studies on dynamic navigation have been published to date [26,37,38,39,40,41,42,43,44,45]. The published studies show heterogeneous factors such as different dynamic navigation systems, different implant planning programs and different implants used. A mean deviation in the angular deviation of 2.26 ± 1.62° [42] to 6.46 ± 3.95° [41] with a calculated unweighted average of 3.67° was found [26,37,38,39,40,41,42,43,44,45]. The 3D deviation at the implant emergence point varied between 0.67 ± 0.29 mm [44] and 1.37 ± 0.55 mm [39]. An unweighted mean value of 1.03 mm was calculated [26,37,38,39,40,41,42,43,44,45]. The published data on the accuracy of dynamic navigation have so far been analyzed in a few systematic reviews [32,36,46,47,48]. The implant emergence point is particularly important for a predictable prosthetic result and has a significant influence on the esthetic outcome [49]. The values of this study are at the upper end of accuracy compared to previous studies on dynamic navigation. 

The mean 3D deviation at the implant apex for group A was 3.0 mm (95% CI 2.3–3.7 mm), and for group B, it was 1.8 mm (95% CI 1.3–2.3 mm). Consideration of these deviations is essential for the planned safety margin. A safety distance of at least 2.0 mm is specified for static navigation. Group B shows that this distance must also be maintained for dynamic navigation [7,46]. A higher safety distance would have to be selected for group A. The dynamic navigation system used in this study still needs to be tested for its clinical reliability in further studies in order to be able to specify the required safety distances. Prior to this, it should not be used in cases of extremely low bone availability or transgingival surgery in order to avoid damaging neighboring anatomical structures.

Group A showed a mean angular deviation of 6.3° (95% CI 4.0–8.7°), and group B showed an angular deviation of 2.7° (95% CI 1.6–3.9°). The values of group B show a similar precision as in most previously published studies [32]. Group A shows significantly less precise results. In their study, Jorba-Garcia et al. reported a mean angular deviation of 3.68° (95% CI 3.61–3.74°) [46], and Wei et al. reported a mean angular deviation of 4.22° (95% CI 2.74–5.68°) [47]. In further studies, values of around 3.807° (95% CI 3.083–4.530) were achieved [48]. With inclined implant axes, the use of customized abutments is essential for the correct design of the proximal contacts.

The analysis of the accuracy by jaw and process chain should show whether this has an influence on the accuracy between the planned and actual implant positions. The mean value calculated from the angular deviations of the inserted implants in the lower jaw is 5.35°. In the upper jaw, the mean value is 2.3°. This difference may be due to the lower number of implants inserted in the upper jaw. Other influencing factors in the lower jaw, such as (for example) tongue pressure, are conceivable. The mouth opening, the restricted view of the surgical area in the posterior region and the movements of the patient, for example, can be possible influencing factors in implant placement with a dynamic navigation system [27]. The mean value of the average 3D deviation at the implant shoulder is 2.2 mm in the mandible and 1.6 mm in the maxilla. No significant differences in the accuracy of the 3D deviation can be determined. Due to the small number of test subjects, these values can only be classified as informative. Further clinical studies are necessary to verify the trend of the results. In three subjects, the reference marker could not be recognized intraoperatively by the tracking system, and the implantation could not take place as planned. All these cases were from process chain A. Two cases with planned implantations in the maxilla and one case in the mandible were affected. In each of the failed detections of the reference markers in the maxilla, implants in region 26 were to be inserted, i.e., the reference marker was positioned in the posterior region of the contralateral side. In order to establish a sufficient connection between the optical tracking unit on the surgical handpiece and the reference marker, the surgeon has to make an unusual rotation of the hand in this case. This is mainly due to the common setup of a clinical environment during dental surgery, in which the surgeon is positioned to the right of the patient. 

Overall, it can be described that the intraoperative handling in procedure A appears to be clearly prone to errors. The marker tray with the thermal impression material does not always sit perfectly and reproducibly on the contralateral residual dentition, and the positioning must be continuously checked. In particular, patients with little residual dentition and teeth with little undercut could be a limiting factor for this procedure. In addition, the marker tray can easily be displaced by tongue and cheek pressure if it is not fixed correctly. A residual dentition is absolutely necessary for the dynamic navigation system investigated, which rules out implant placement in an edentulous jaw. Positioning on an edentulous jaw with transversal pins or screws would be conceivable, but this excludes a procedure according to workflow A. Furthermore, the surgeon and assistant must be a well-coordinated team so that the optical connection between the reference marker and the tracking system is not interrupted during the operation.

In this study, the difference in accuracy between the planned and actual implant positions was evaluated using a postoperative intraoral scan. Van Hooft et al. showed in their study that the use of a postoperative intraoral scan for evaluation is similarly accurate to the use of a CBCT. The deviations are only of minimal clinical significance. The reduction in radiation when using a postoperative scan is extremely relevant for the patient [50]. Further studies have shown that evaluation using an intraoral scan does not lead to significantly less accurate results [51,52]. An intraoral scan for evaluation can be classified as significantly more precise than a CBCT as the starting point for implant planning. If more precise digitalization is used to determine the deviation of the implant position, this has an influence on the results obtained. The major technical advantage of an intraoral scan is the possibility of recording both the implant position achieved and the clinical patient situation in one data set. The accuracy of the evaluation can also be influenced by blurring artifacts in the CBCT. These artifacts are caused by patient movements during the scan, which last several seconds [53].

The implant system itself can have an influence on accuracy [54] in addition to the influencing factors already mentioned, which have been investigated in studies on static navigation [55]. In their retrospective study, Wu et al. found no significant differences between dynamic and static navigation in terms of 3D deviation at the implant shoulder and the implant apex and in terms of angular deviation when evaluating 95 implants [26]. Similar results were obtained in a clinical study by Yimarj et al. [45]. Jaemsuwan et al. were able to show in their study that dynamic navigation shows no significant differences in terms of accuracy compared to static navigation. However, statically and dynamically navigated implantation is significantly more accurate than freehand implantation [35]. The accuracy values achieved in this study were slightly worse than those of static navigation, which can presumably be attributed to the practically freehand preparation of the implant bed. In static navigation, guidance takes place via the sleeves. This study shows that high precision can be achieved when placing implants using dynamic navigation. However, it also shows that significant differences can occur even with minor variations in the referencing method. This is also confirmed by studies and comparisons of different navigation systems, some of which have significantly different referencing methods [34,56]. Analogies between individual systems are therefore not appropriate. The clinical applicability and precision of each system must be examined and verified. With the different systems, errors due to patient movement are conceivable when referencing the drill guide during implantation. Intra-oral fixation of the marker can rule out movement artifacts.

The small number of test subjects (30 patients) was one of the greatest limitations of this study, which is why it should be considered a pilot study. A larger number of subjects would enable a more detailed evaluation of various parameters and, thus, a more well-founded analysis. an evaluation according to jaw, tooth position and the number of remaining teeth can only be assessed to a limited extent in this study. Further parameters, such as the comparison between flapless implantation and implantation with the formation of a periosteal flap, could be investigated in studies with a larger number of subjects. In addition, only one implant system was used, and all implants were placed by the same experienced surgeon. Therefore, future studies using different implant systems and having implants placed by different surgeons are needed.

## Figures and Tables

**Figure 1 bioengineering-11-00155-f001:**
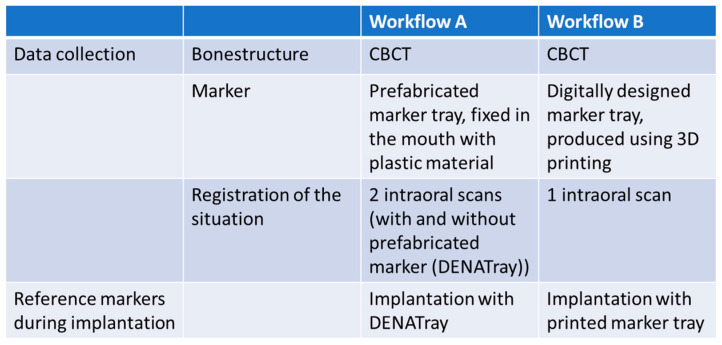
Illustration of the two workflows for the application of the reference markers and the clinical implementation.

**Figure 2 bioengineering-11-00155-f002:**
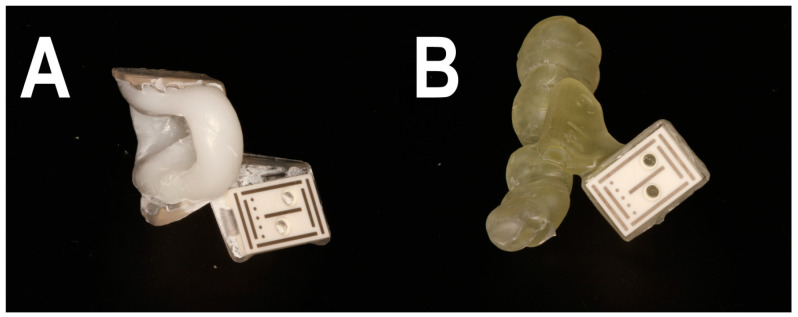
(**A**) Prefabricated marker holder (DENATray) with plastic impression material (**B**) Designed and printed marker tray with inserted reference marker for positioning on the contralateral side during implantation.

**Figure 3 bioengineering-11-00155-f003:**
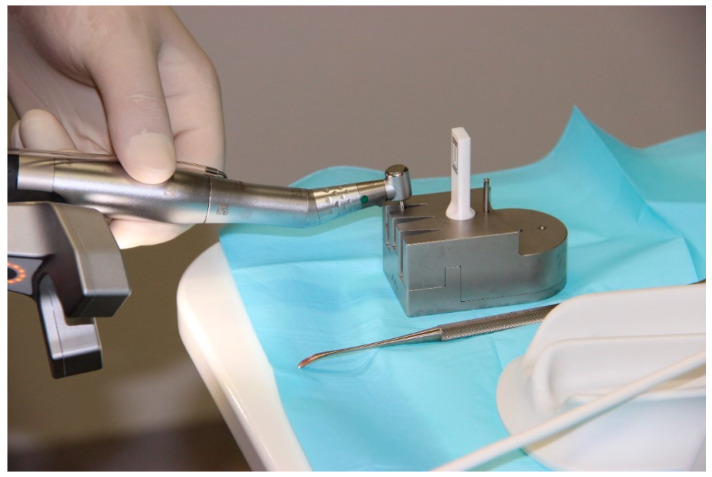
Registration process of the respective drill before each drilling step during implantation. Optical unit mounted on the surgical handpiece.

**Figure 4 bioengineering-11-00155-f004:**
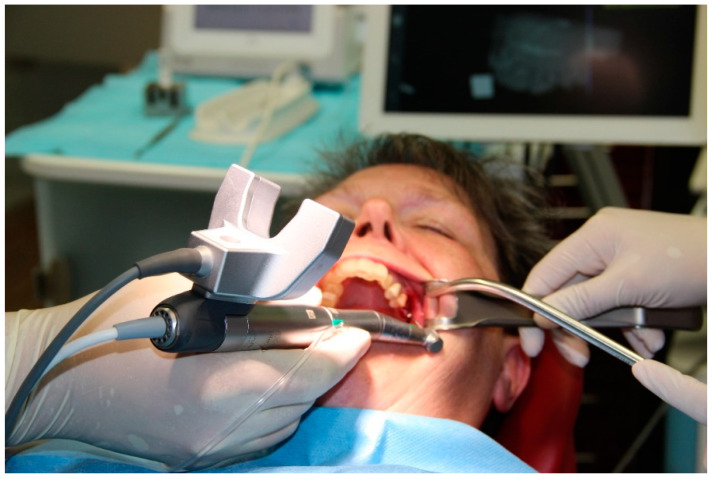
Optical unit positioned on the surgical contra-angle handpiece during implantation and screen of the dynamic navigation system in the background for interactive guidance of the surgeon.

**Figure 5 bioengineering-11-00155-f005:**
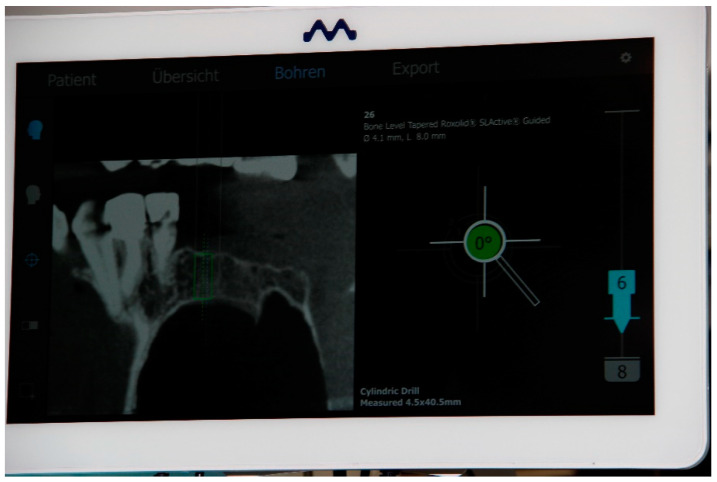
Three-dimensional display on the screen of the dynamic navigation system during implantation to control height, depth and angle in real time. Left: representation of the planning in a section of the patient’s digital volume tomography. Angular deviation (green) in degrees (°). Depth (turquoise), planned implant with length and diameter and calibrated drill in millimeters (mm).

**Figure 6 bioengineering-11-00155-f006:**
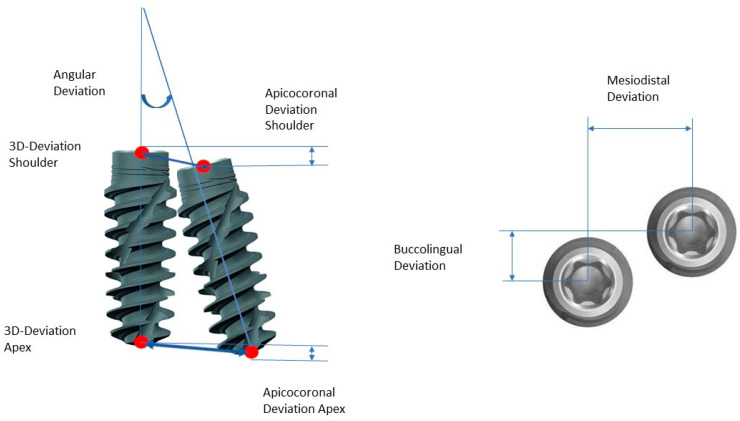
Analysis of the implant positions with all measurements taken. Implant center points at the implant shoulder and apex (red).

**Figure 7 bioengineering-11-00155-f007:**
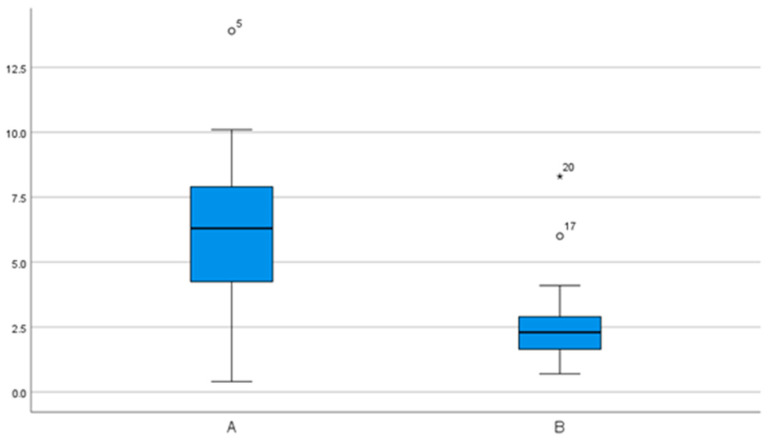
Angular deviations in degrees (°) in group comparison. Significantly more accurate values for group B.

**Figure 8 bioengineering-11-00155-f008:**
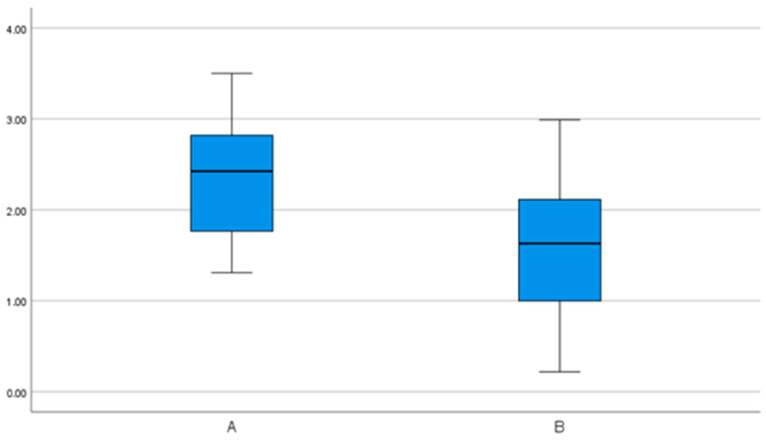
Three-dimensional (3D) deviation at the implant shoulder in millimeters (mm) in group comparison. Significantly higher accuracy in group B.

**Figure 9 bioengineering-11-00155-f009:**
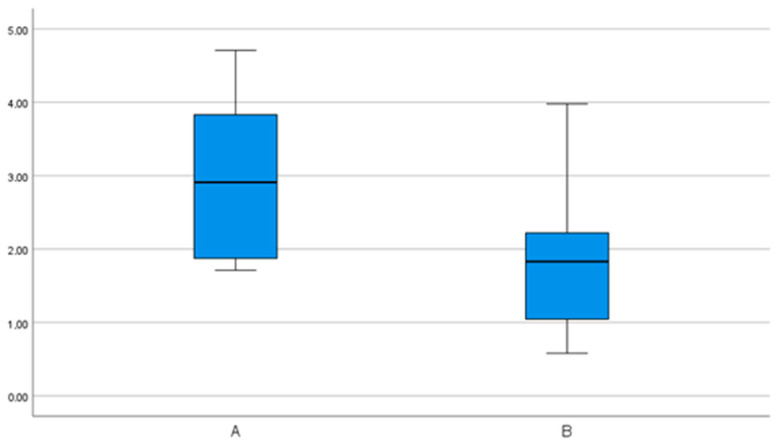
Three-dimensional (3D) deviation at the implant apex in millimeters (mm) in group comparison. Significantly more accurate values for group B.

**Table 1 bioengineering-11-00155-t001:** Demographic distribution of recruited patients.

Group Distribution (Number of Patients)		Group A	12
		Group B	15
Group distribution (number of patients)	Group A	Women	7
		Men	5
	Group B	Women	7
		Men	8
Group distribution (number of patients)		20–30 years	3
		30–40 years	4
		40–50 years	7
		50–60 years	4
		60–70 years	7
		70–80 years	2

**Table 2 bioengineering-11-00155-t002:** Demographic distribution of inserted implants, implant position, distribution across the jaws and different implants.

Positions of the implants (number of each position)	Regio 16	1
	Regio 23	1
	Regio 24	1
	Regio 25	6
	Regio 35	1
	Regio 36	7
	Regio 45	4
	Regio 46	6
Distribution of upper/lower jaw (number of respective jaws)	OK	9
	UK	18
Implant distribution (number of implants in each case)	Straumann BLT Length 8.0 mm Diameter 3.3 mm	2
	Straumann BLT Length 8.0 mm Diameter 4.1 mm	2
	Straumann BLT Length 10.0 mm Diameter 3.3 mm	7
	Straumann BLT Length 10.0 mm Diameter 4.1 mm	11
	Straumann BLT Length 10.0 mm Diameter 4.8 mm	2
	Straumann BLT Length 12.0 mm Diameter 3.3 mm	1
	Straumann BLT Length 12.0 mm Diameter 4.1 mm	2

**Table 3 bioengineering-11-00155-t003:** Deviations of the achieved implant position from the planned implant position. Evaluation of the 2 clinical process chains for the number (N) of patients and inserted implants. Specification of the mean values, the standard deviation (SD), the 95 percent (%) confidence interval (CI), the minimum (Min) and maximum (Max) values for the deviations at the implant shoulder and the implant apex in three-dimensional (3D), mesiodistal, buccolingual and apicocoronal directions in millimeters (mm) and the angular deviation in degrees (°).

	Group A N = 12 Patients/12 Implants			Group B N = 15 Patients/15 Implants		
	Medium (SD)	95% CI	Min– Max	Medium (SD)	95% CI	Min– Max
Deviations at the implant shoulder (mm)						
3D	2.35 (0.7)	1.92– 2.78	1.31– 3.5	1.62 (0.8)	1.2– 2.05	0.22– 2.99
Mesiodistal	0.17 (1.7)	−0.9– 1.2	−2.8– 2.7	0.6 (0.8)	0.2– 1.04	−0.6– 2.4
Buccolingual	−0.1 (1.0)	−0.8– 0.5	−1.3– 1.8	−0.5 (0.7)	−0.93– −0.1	−2.0– 0.9
Apicocoronal	−1.3 (0.9)	−1.9– −0.8	−2.3– 1.0	−0.74 (1.0)	−1.3– −0.2	−2.1– 0.8
Deviations at the implant tip (mm)						
3D	3.0 (1.1)	2.3– 3.7	1.7– 4.7	1.8 (0.9)	1.3– 2.3	0.6– 4.0
Mesiodistal	0.1 (2.5)	−1.6– 1.7	−3.4– 4.3	0.7 (1.1)	0.1– 1.24	−0.7– 3.3
Buccolingual	−0.3 (1.5)	−1.2– 0.7	−2.9– 2.2	−0.6 (0.8)	−1.1– −0.1	−2.1– 0.7
Apicocoronal	−1.3 (0.8)	−1.8– −0.7	−2.3– 1.0	−0.7 (1.0)	−1.3– −0.2	−2.1– 0.8
Angular deviations (°)	6.3 (3.7)	4.0– 8.7	0.4– 13.9	2.7 (2.0)	1.6– 3.9	0.7– 8.3

## Data Availability

The data presented in this study are available from the authors upon reasonable request.

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
