# Peer review of "Accuracy of Dental Implant Placement with Dynamic Navigation—Investigation of the Influence of Two Different Optical Reference Systems: A Randomized Clinical Trial"

_bioengineering, 2024, doi:10.3390/bioengineering11020155_

Round 1
Reviewer 1 Report
Comments and Suggestions for Authors
Very nicely done and explained clinical experimental study ! Congratulations !
Author Response
Our team was very pleased to receive your comment - thank you very much.
Reviewer 2 Report
Comments and Suggestions for Authors
This clinical study provides more clinical evidence for implant navigation surgery. It compares the accuracy of two workflow methods for Mininavident navigator, which is somewhat innovative. However, this study was limited to scenarios in which this specific navigation system was used, so it may have little application significance. Here are some suggestions for revision.
1. The language is well-organized and coherent.
2. It is suggested that the introduction section should focus more on the background of this study. The background of the two workflows involved in this study should be emphasized, and the content related to the guide plate can be introduced more briefly.
3. In order to let readers clearly understand the research content and research purpose of this paper, it is suggested that the author briefly introduce the two workflow methods at the end of the introduction section, and specifically mention the two methods in the purpose of this study instead of using pronouns.
4. How was the sample size determined? Authors are advised to add more details about this at the beginning of the Methods section.
5. It would be better to provide a flow diagram of workflow A and workflow B.
6. It is recommended to provide an image of the marker used in workflow A.
7. It is recommended to begin the discussion section by responding to the null hypothesis of this paper.
8. Please further discuss the clinical implications of the conclusions of this paper. Different navigation systems have different marker fixation devices. Are the conclusions of this article instructive for other navigation system guided implant surgery? It is suggested that the authors cite more literatures for analysis.
9. The format of "positions of the implants" in Table 2 needs to be adjusted.
Reviewer 3 Report
Comments and Suggestions for Authors
Dear Authors,
I consider your manuscript a great work!I suggest some changes to improve from some points of view.
The paper is a randomized clinical trial on the accuracy of dental implant placement using dynamic navigation.
The Authors made a great work in terms of methodology and the paper sounds scientific and well written.
However some improvements are mandatory before acceptance.
As regards the title, I suggest changing it so that the "type of study" and therefore "A randomized clinical trial" is positioned at the end of the title, after ":". The abstract is well written, complete and summary in its various aspects. The keywords are complete and appropriate.
Too many double spaces were found during the revision: please fix it.
In the introduction, which is well written and complete, I suggest some revisions to the Authors:
· in particular, "Many studies show a higher accuracy of navigated implant placement compared to the conventional freehand method." If this is correct regarding freehand surgery, what if we compare it with a static guided surgery? What differences can be assessed?
· In the introduction, I suggest also considering the aspect of durability also in terms of hygienic maintenance of the prosthetic restorations that are delivered on the implants. In particular, a prosthesis with large undercuts, difficult to clean, and often cemented represents a limiting factor in maintenance which can favor the compromise of rehabilitation, as underlined by "Reda R, Zanza A, Cicconetti A, Bhandi S, Guarnieri R, Testarelli L, Di Nardo D. A Systematic Review of Cementation Techniques to Minimize Cement Excess in Cement-Retained Implant Restorations. Methods Protoc. 2022 Jan 17;5(1):9. doi: 10.3390/mps5010009."
Materials and methods are clear and well explained. Different aspects are analyzed with a dedicated statistical test. The authors did a great job in the explication of all the variables identified and included in the study.
Results are easy to understand and comprehensive. All the studied characteristics were reported in tables which are clear and concise.
Discussion: the overall is comprehensive, concise and complete in its various aspects. I suggest comparing the data found by the Authors in the present study with other manuscripts currently present in the literature on this interesting topic.
In the discussion:
· “movements of the patient” How can patient movements affect implant insertion? During site preparation, is the position of the patient's head in space fundamentally entrusted to the optical reader and matching software? and, in the lower arch, where in addition to the movement of the patient's head in space, there is also the movement of the jaw in space, shouldn't we find a greater discrepancy?
I suggest that the authors include a short section of conclusions to summarize what has been described in the discussions and highlight the salient aspects of the manuscript.
Bibliography should be formatted respecting the journal’s requirements and no improper citations are evidenced.
Figures and labels are clear and easy to comprehend.
English is clear and easy to understand.
Author Response
Many thanks for the comments. Our comments and the revisions are attached

Round 2
Reviewer 3 Report
Comments and Suggestions for Authors
Dear Auhtors,
I believe the article has been edited correctly and is now suitable for publication. I believe it is only necessary to review the style of the biography according to MDPI's indications.
Author Response
We are happy to resubmit the manuscript.
We have once again checked the list of titles against the author guidelines. Apart from a simple change in the layout format, we did not notice any changes. We kindly ask you to check it.
We are pleased that the manuscript has passed the review process.